# Research on Factors Affecting Mine Wall Stability in Isolated Pillar Mining in Deep Mines

**Jiang Guo** [1] , **Xin Cheng** [1], **Junji Lu** [2], **Yan Zhao** [1,*] and **Xuebin Xie** [1]

1   School of Resources and Safety Engineering, Central South University, Changsha 410083, China; guojiang@csu.edu.cn (J.G.); chengxin129@163.com (X.C.); xbxie@csu.edu.cn (X.X.)
2   Hunan Lianshao Construction Engineering (Group) Co., Ltd., Changsha 410001, China; lujjlujj780902@126.com
*   Correspondence: zyzhaoyan@csu.edu.cn

**Abstract:** This study takes the Dongguashan Copper Mine as its engineering background. Based on the mechanical model of the mine wall under the trapezoidal load of the backfill, a comprehensive evaluation index is proposed, and its calculation equation is derived. On this basis, an orthogonal test is designed to explore the influence of mining design parameters on mine wall stability. The results show that the width of the mine wall is the main factor affecting its stability, and increasing the width of the mine wall can significantly improve its stability. When the width of the mine wall is kept above 4 m, its stability is better. When the mechanical parameters of the backfill are poor, the mine wall is prone to overturning failure. The width of the mine room has an influence on the multi-directional loading of the mine wall, but the influence on the stability of the mine wall is low. According to the regression equation calculation, the mine wall safety factor is about 1.46 under the design of G5 mining of Dongguashan Line 52, the stability of the mine wall is good after actual mining and the engineering application effect is ideal, which can provide a theoretical basis for the design of isolation pillar mining in deep mines.

**Keywords:** mine wall stability; Platts arch; safety factor; orthogonal experiment; regression analysis



## 1. Introduction

With the continuous development and utilization of resources, the mining of mineral resources has developed towards a deeper level and a larger span, and deep mining has become the mainstream trend in mining [1–6]. As a mining method, fill mining can make full use of tailings resources and control the deformation of surrounding rock effectively, and it is widely used in deep mining [7,8]. After the goaf is filled, the backfill body can support the rock formation and control the stope pressure activities. In order to improve the utilization rate of mineral resources, the ore pillars in the stope often need to be mined after the goaf is filled. During the mining of the ore pillar, in order to ensure the stability of the backfill and the goaf, mine walls are often left on both sides of the pillar. The wall can improve the stability of the backfill and play a temporary supporting role to avoid the collapse of the backfill and the falling of the roof during the mining of the pillar, which increase dilution losses from mining techniques. Therefore, it is of great significance to carry out research on the stability of the mine wall and optimize the structural parameters of the mine wall, which is of great significance to reduce the loss of mine wall ore volume and ensure the safety of mining pillar.

At present, many scholars have achieved a lot of research results through theoretical calculation, reliability analysis and numerical simulation [9–13]. Elasticity theory, thin plate theory and catastrophe theory have been widely used in the study of mine wall stability. Wei [14] improved the limit equilibrium method by applying the stress area superposition method, in which the stress distribution of the elastic area and the inelastic area of the pillar was obtained and the stability of the pillar in the strip-mining process was analyzed.

Huang [15] established a mechanical model of mine wall bending failure; the critical bending stress of the mine wall was calculated by the energy method and the stability changes in its size were obtained and the correctness of the theoretical analysis was verified by numerical simulation. Xu [16] proposed the concept of stripping degree and established a corresponding secondary stripping model to analyze the stability of hyperbolic coal pillars. He considered the critical condition of pillar instability under the influence of coal pillars and high temperatures. Based on the instability theory and cusp catastrophe model, Wang [17] deduced the roof deflection curve under the condition of the unequal span of adjacent stopes, and then the pillar instability condition under the condition of asymmetric mining was determined. Based on reliability analysis, Idris [18] introduced the method of using an artificial neural network in pillar stability research. The pillar reliability index and failure probability were calculated, and he analyzed the influence of variation coefficient on pillar stability and determined it according to the minimum acceptable risk of pillar failure. The optimal pillar size was determined. Liu [19] studied the failure mechanism of the pillar group by using the method of the renormalization group, and the safety factor of the pillar system was obtained. The safety of the pillar group scheme was analyzed with the fuzzy comprehensive evaluation theory, and the pillar structure design in seabed mining was optimized. Ding [20] used the Stochastic Gradient Boosting (SGB) model to analyze pillar stability and established an evaluation index system based on five factors: pillar width, pillar height, pillar aspect ratio, rock uniaxial compressive strength and pillar stress. The parameter sensitivity was studied based on the relative variable importance, and the main variables affecting the pillar stability were obtained. With the development of computers, numerical simulation has also become an important means of research on pillar stability. Zhang [21] studied the relationship between various factors in multi-coal strip mining and the vertical displacement of the pillar based on FLAC numerical simulation software. Zhang [22] used the layered cover model for pillar design and stability analysis, eliminating the influence of boundary effects on the analysis results, and determined the main factors affecting pillar stability. Yang [23] combined the experimental data and the research method of numerical simulation, studied the pillar size of the Zhaozhuang Coal Mine in Shanxi Province, and proposed differentiated support technology, which improved the bearing strength of the pillar and effectively reduced the deformation of the surrounding rock.

In general, there are many research methods for mine wall stability analysis, and research results have also been obtained. Among them, reliability analysis can comprehensively consider the influence of various factors on the stability of the mine wall, and it is widely used in engineering. However, the current stability and reliability analysis of the mine wall is mostly related to unidirectional vertical load; the main consideration is the yield failure form of the mine wall, ignoring the influence of backfill on mine wall stability [24,25]. Furthermore, the backfill method is mostly used for mining in deep mines. As well as the yield failure of the overlying surrounding rock, overturning failure may also occur under the lateral load of the backfill, so existing research results are not suitable for the mine wall design of the backfill stope. Therefore, taking the Dongguashan Copper Mine as an example, this study analyzes the backfill-wall bearing mechanism after the isolated pillar is mined. The expression of the safety factor of the mine wall under the action of overlying strata and backfill is deduced, and orthogonal experiments are designed to study the influence of different parameters on the stability of the mine wall. The equation for calculating the comprehensive safety factor of the mine wall is obtained. The research results are expected to provide a reference for the design of isolated pillar mining in deep mines.

## 2. Project Overview

The Dongguashan Copper Mine is located in Tongling, Anhui Province. It is a typical deep deposit with a burial depth of $-690$ to $-1000$ m, a main ore body of 1810 m in length, 500 m in width and an average thickness of 20 to 50 m. The main ore body of the Dongguashan Copper Mine strikes NE35$-$40$°$, the average dip angle is 20$°$, and the maximum dip angle is located on the two sides of the ore body, about 30–40$°$. The middle

part of the ore body is relatively gentle, and the lateral angle is less than 15°. The ore-bearing rock mass is mainly copper-bearing skarn and copper-bearing magnetite. The roof-surrounding rock has good properties; the surrounding rock is mainly composed of marble, and the floor surrounding rock is mainly composed of siltstone and diorite.

Due to the deep burial of the ore deposit and the high stress state of the surrounding rock, the mining period of the ore rock is long and the mining is difficult. In order to achieve the goals of efficient, safe, and low-cost mining, the main ore body adopts the "temporary isolation of the pillar stage and the subsequent filling mining method". The method is characterized by dividing the panel along the strike direction of the ore body; the panel span is 100 m, and the panel size is length (ore body width) × width (100 m) × height (ore body thickness), and there are temporary isolation pillars in each panel interval to achieve independent mining of each panel. The stopes are divided every 18 m along the vertical ore body strike (the width direction of the ore body) inside each panel. The mining process is carried out in three steps. First, the one-step stopes are mined according to the method of "interval mining". After the one-step stope is mined, full tailings are used for cementing and filling. After the first-step stopes are filled, the stopes are recovered and filled in a two-step process. At this stage, the one-step and two-step stopes' mining work has been completed. In order to fully recover the mineral resources, it is necessary to carry out three-step mining for the isolation pillar between the panel. After the isolation pillar is mined, mine walls are left on both sides to prevent a large number of backfill in the stope from collapsing. Figure 1a is a top view of the stope layout and Figure 1b is a vertical cross-section view of the stope layout.

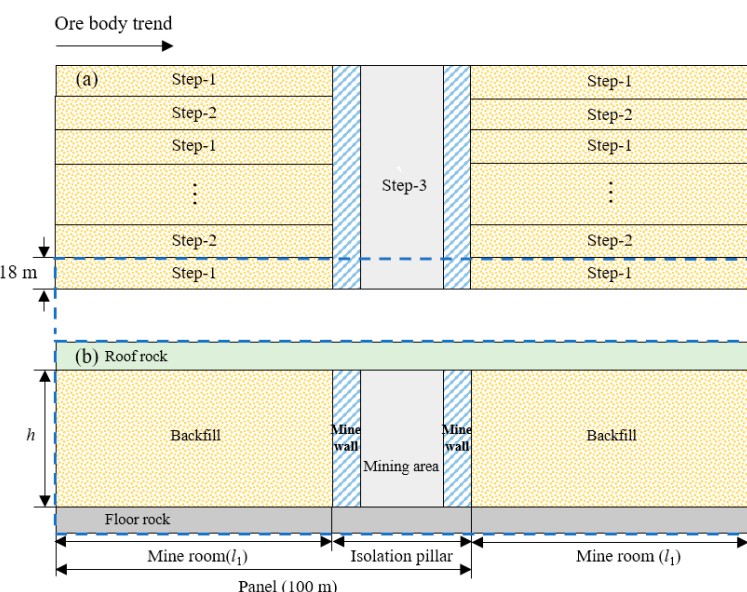

**Figure 1.** (**a**) Top view of Dongguashan stope layout. (**b**) Vertical cross section view of Dongguashan stope layout.

## 3. Analysis of Mine Wall Safety Factor

Mine wall instability includes various failure forms, among which are mainly brittle failure, ductile failure and weak plane shear failure [26,27]. The mine wall is simultaneously subjected to the load of the overlying surrounding rock and the backfill, which may cause brittle compression failure under the action of the overlying surrounding rock or shear failure under the lateral load of the backfill [28–31]. In addition to the above-mentioned strength failure, the mine wall may also suffer from overturning failure. Combined with the actual situation of Dongguashan, this study takes the comprehensive safety factor as an evaluation index of mine wall safety. Among them, when the safety factor value is greater than 1, it means that the mine wall is in a stable state. When the safety factor is equal to

1, it means that the mine wall is in a critical state. When the safety factor is less than 1, it means that the mine wall is in an unstable state.

### 3.1. Mine Wall Mechanics Model

According to Platts' ground pressure theory, when excavation is carried out, the original initial stress state of the surrounding rock will be destroyed, and the stress will be redistributed [29]. When the stress field becomes stable again, a pressure arch containing the plastic zone will be formed, namely the Platts arch [30]. After the mining room is mined, the self-weight of the surrounding rock in the upper arched plastic area is borne by the backfill, which is denoted as $G_1$. Taking the isolated ore pillar as the research object, when the isolated ore pillar is mined, the rock body above the goaf will cave in, and a temporary stable caving arch will gradually form. The self-weight of the surrounding rock of the caving arch is $G_2$. Figure 2a shows the bearing mechanism diagram of the backfill–mine wall system. Considering the backfill as a homogeneous medium, the effect of the backfill body on the mine wall increases linearly along the depth direction. When the backfill bears the direct upper load, the initial value of the top load on the backfill is not 0, and there is an initial lateral effect on the top of the mine wall. This study considers the force form of the mine wall under the trapezoidal lateral load. The mechanical model of the mine wall is shown in Figure 2b.

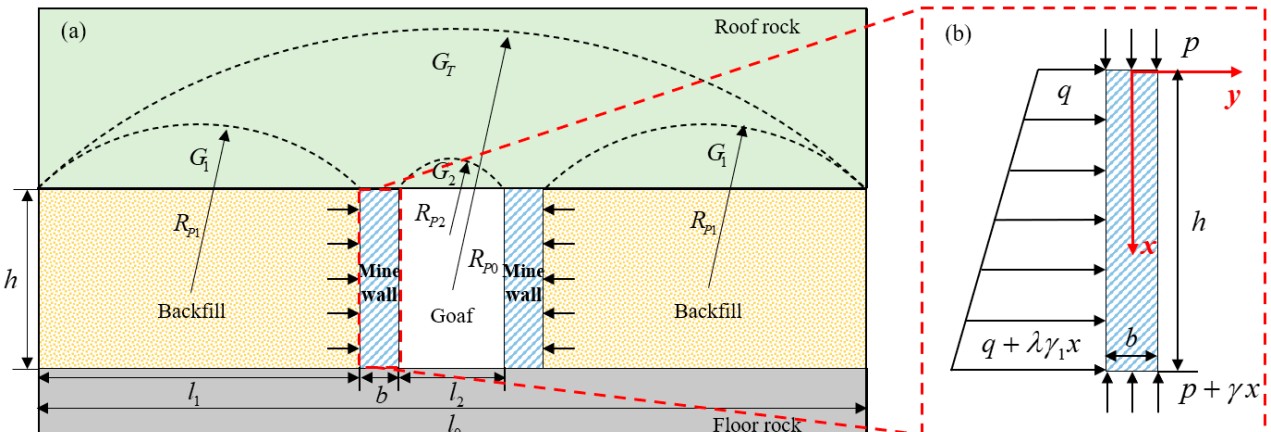

**Figure 2.** (**a**) Bearing mechanism diagram of the backfill–mine wall system. (**b**) Mine wall mechanic model.

In Figure 2a: $G_T$ is the total weight of the overlying surrounding rock of the backfill–mine wall system; $G_1$ is the self-weight of the upper surrounding rock for the backfill; $G_2$ is the self-weight of the caving surrounding rock in the isolated pillar goaf; $R_{P0}$ is the equivalent radius of the plastic zone of the backfill–mine wall system; $R_{P1}$ is the equivalent radius of the upper plastic zone of the backfill; $R_{P2}$ is the equivalent radius of the plastic zone in the upper part of the goaf; $l_0$ is the span of the backfill–mine wall system; $l_1$ is the width of the mine room; $l_2$ is the span of the goaf.

In Figure 2b: $p$ is the initial upper load; $q$ is the initial lateral load; $\lambda$ is the lateral pressure coefficient; $\gamma_1$ is the bulk density of the backfill; $\gamma$ is the bulk density of the mine wall; $h$ is the height of the mine wall; $b$ is the width of the mine wall, $x$ is the vertical distance from the origin of the coordinates, $y$ is the lateral distance to the origin of the coordinates.

According to our previous research [31–33], the stress expression of the mine wall under trapezoidal load was obtained by the semi-inverse solution method of elastic mechanics:

$$
\begin{cases}
\sigma_x = -\dfrac{2\lambda\gamma_1 y}{b^3}x^3 - \dfrac{6qy}{b^3}x^2 + \left(\dfrac{4\lambda\gamma_1 y^3}{b^3} + \dfrac{6qhy}{b^3} - \dfrac{3\lambda\gamma_1 y}{5b} + \dfrac{6\gamma y}{b}\right)x \\
\qquad + \dfrac{4qy^3}{b^3} + \dfrac{6qhy}{b^3} - \dfrac{3qhy}{5b} - p - \gamma x \\
\sigma_y = \left(-\dfrac{2q}{b^3} - \dfrac{2\lambda\gamma_1 x}{b^3}\right)y^3 + \left(\dfrac{3q}{2b} + \dfrac{3\lambda\gamma_1 x}{2b}\right)y - \dfrac{\lambda\gamma_1 x}{2} - \dfrac{q}{2} \\
\tau_{xy} = \left(\dfrac{3\lambda\gamma_1}{4b} - \dfrac{3\lambda\gamma_1 y^2}{b^3}\right)x^2 + \left(-\dfrac{3q}{2b} - \dfrac{6qy^2}{b^3}\right)x + \lambda\gamma_1\left(\dfrac{b}{80} - \dfrac{3y^2}{10b} + \dfrac{y^4}{b^4}\right) \\
\qquad - \gamma_1\left(\dfrac{b}{4} + y - \dfrac{3y^2}{b}\right) - \dfrac{3qh}{4b} - \dfrac{3qhy^2}{b^3}
\end{cases}
\tag{1}
$$

In Equation (1): $\sigma_x$ is the vertical stress of the mine wall, $\sigma_y$ is the lateral stress of the mine wall, $\tau_{xy}$ is the shear stress of the mine wall.

According to Platts' theory, the plastic zone radius correction coefficient $\xi$ is [26,27]:

$$
\xi = \left[\frac{(\gamma_2 H_0 + C_t \cos\varphi_1)(1 + \sin\varphi_1)}{C_t \cos\varphi_1}\right]^{\frac{1-\sin\varphi_1}{2\sin\varphi_1}}
\tag{2}
$$

The dead weights of the backfill–mine wall system, the backfill, and the caving surrounding rock in the goaf are, respectively,

$$
G_T = \left[\sqrt{\left(\frac{h}{2}\right)^2 + \left(\frac{l_0}{2}\right)^2}\,\xi - \frac{h}{2}\right]l_0\gamma_2
\tag{3}
$$

$$
G_1 = \left[\sqrt{\left(\frac{h}{2}\right)^2 + \left(\frac{l_1}{2}\right)^2}\,\xi - \frac{h}{2}\right]l_1\gamma_2
\tag{4}
$$

$$
G_2 = \left[\sqrt{\left(\frac{h}{2}\right)^2 + \left(\frac{l_2}{2}\right)^2}\,\xi - \frac{h}{2}\right]l_2\gamma_2
\tag{5}
$$

Then, the initial vertical load of the mine wall $p$ is:

$$
p = \frac{G_T - 2G_1 - G_2}{Nb}
\tag{6}
$$

The initial lateral load $q$ is calculated by the Rankine earth pressure equation:

$$
q = \frac{\lambda G_1}{l_1} = \tan^2\left(45 - \frac{\theta}{2}\right)\left(\sqrt{\left(\frac{h}{2}\right)^2 + \left(\frac{l_1}{2}\right)^2}\,\xi - \frac{h}{2}\right)\gamma_2
\tag{7}
$$

where $\gamma_2$ is the bulk density of the roof rock; $H_0$ is the mining depth; $C_t$ is the cohesion of the roof rock; $\varphi_1$ is the friction angle of the roof rock; $N$ is the number of pressure-bearing mine walls; $\theta$ is the friction angle of the backfill.

### 3.2. Compression Safety Factor $K_\sigma$

The compressive safety factor represents the compressive capacity of the mine wall, and its calculation equation is the ratio of the compressive strength of the mine wall to the maximum compressive stress on the mine wall. The maximum compressive stress on the mine wall is the maximum value of $\sigma$ in Equation (1). For the value of the compressive strength of the mine wall, some scholars have deduced a variety of calculation equations for the bearing strength of the mine wall. Among them, the empirical equation proposed

by Bieniawski [34] is more accurate and widely used. The equation for calculating the compressive strength of the mine wall is:

$$\sigma_p = \sigma_c \left[ 0.64 + 0.36 \left( \frac{b}{h} \right) \right]^\alpha \tag{8}$$

In Equation (8): $\sigma_p$ represents the compressive strength of the mine wall; $\sigma_c$ represents the uniaxial compressive strength of the complete rock mass; $b$ represents the width of the mine wall; $h$ represents the height of the mine wall; $\alpha$ is a constant, and its value depends on the size of the mine wall. When the ratio of the width to height dimension of the mine wall is greater than 5, the value of $\alpha$ is 1.4, and when the aspect ratio is less than 5, the value of $\alpha$ is 1.0.

Therefore, the calculation equation of the mine wall compression safety factor is:

$$K_\sigma = \frac{\sigma_p}{\max(\sigma)} = \frac{\sigma_c \left[ 0.64 + 0.36 \left( \frac{b}{h} \right) \right]^\alpha}{\max \left\{ \begin{array}{l} -\frac{2\lambda\gamma_1 y}{b^3} x^3 - \frac{6qy}{b^3} x^2 + \left( \frac{4\lambda\gamma_1 y^3}{b^3} + \frac{6qhy}{b^3} - \frac{3\lambda\gamma_1 y}{5b} + \frac{6\gamma y}{b} \right) x \\ +\frac{4qy^3}{b^3} + \frac{6qhy}{b^3} - \frac{3qhy}{5b} - p - \gamma x, \\ \left( -\frac{2q}{b^3} - \frac{2\lambda\gamma_1 x}{b^3} \right) y^3 + \left( \frac{3q}{2b} + \frac{3\lambda\gamma_1 x}{2b} \right) y - \frac{\lambda\gamma_1 x}{2} - \frac{q}{2} \end{array} \right\}} \tag{9}$$

### 3.3. Shear Safety Factor $K_\tau$

The mine wall shear safety factor is an index reflecting the size of the mine wall shear resistance, and its calculation equation is the ratio of the shear strength of the mine wall shear plane to the maximum shear stress on the mine wall. The maximum shear stress on the mine wall is the maximum value of $\tau_{xy}$ in Equation (1). The shear strength of the mine wall can be calculated by the Mohr–Coulomb strength criterion, and its calculation expression is:

$$\tau_p = \sigma_0 \tan \varphi + c \tag{10}$$

where $\sigma_0$ is the normal stress in the normal direction of the shear plane of the mine wall; $\varphi$ is the friction angle of the mine wall; $c$ is the cohesion of the mine wall.

Therefore, the calculation equation of the mine wall shear safety factor is:

$$K_\tau = \frac{\tau_p}{\max(\tau_{xy})} = \frac{\sigma_0 \tan \varphi + c}{\max \left\{ \begin{array}{l} \left( \frac{3\lambda\gamma_1}{4b} - \frac{3\lambda\gamma_1 y^2}{b^3} \right) x^2 + \left( -\frac{3q}{2b} - \frac{6qy^2}{b^3} \right) x + \lambda\gamma_1 \left( \frac{b}{80} - \frac{3y^2}{10b} + \frac{y^4}{b^4} \right) \\ -\gamma_1 \left( \frac{b}{4} + y - \frac{3y^2}{b} \right) - \frac{3qh}{4b} - \frac{3qhy^2}{b^3} \end{array} \right\}} \tag{11}$$

### 3.4. Overturning Safety Factor $K_c$

The overturning safety factor is the anti-overturning ability of the mine wall, and its calculation expression is the ratio of the anti-overturning moment of the mine wall to the force and bending moment of the backfill on the mine wall. According to the force of the mine wall, the expression of the anti-overturning moment of the mine wall is:

$$M_K = \frac{b}{2} (G_p + G) \tag{12}$$

In Equation (12): $M_K$ is the anti-overturning moment of the mine wall; $G_p$ is the load of the overlying surrounding rock on the mine wall and its calculation equation is: $G_p = pb$; $G$ is the gravity of the mine wall per unit thickness, and its calculation equation is: $G = \gamma hb$.

The force of the backfill on the mine wall can be calculated by Rankine earth pressure Equation (13):

$$F_y = qh + \frac{1}{2} \lambda \gamma_1 h^2 \tag{13}$$

The pressure of the backfill on the side of the mine wall is linearly distributed in a trapezoid shape. According to the moment calculation, the action point can be obtained $\frac{3qh+\lambda\gamma_1 h^2}{3(2q+\lambda\gamma h)}$ from the bottom surface, so the overturning safety factor of the mine wall is:

$$K_c = \frac{\frac{b}{2}(G_p + G)}{\left(qh + \frac{1}{2}\lambda\gamma_1 h^2\right)\frac{3qh+\lambda\gamma_1 h^2}{3(2q+\lambda\gamma h)}} \tag{14}$$

*3.5. Comprehensive Safety Factor K*

Considering the various failure forms of the mine wall, the comprehensive safety factor is used as an index to evaluate the stability of the mine wall, and its calculation equation is as follows:

$$K = \min\{K_\sigma, K_\tau, K_c\} \tag{15}$$

In Equation (15): $K$ is the comprehensive safety factor, when $K = K_\sigma$, the main failure form of the mine wall is compression failure; when $K = K_\tau$, the main failure form of the mine wall is shear failure; when $K = K_c$, the main failure form of the mine wall is overturning failure.

## 4. Analysis of Factors Affecting Mine Wall Stability

According to Equation (15), the factors affecting the stability of the mine wall mainly include the width of the mine wall, the height of the mine wall, the friction angle of the backfill, the bulk density of the backfill, the width of the mine room, the mining depth, the uniaxial compressive strength of the mine wall and the mechanical properties of the surrounding rock. Combined with the current mining situation of the Dongguashan Copper Mine. Since the mining depth, mine wall height, mine wall uniaxial compressive strength and mechanical properties of the overlying surrounding rock are determined by the mine geological conditions, it is difficult to artificially design and adjust, so the main consideration of mine wall stability is the width of the mine room, the width of the mine wall, the friction angle of the backfill and the bulk density of the backfill.

According to the measured data of the mine, the mining depth $H_0 = 700$ m, the uniaxial compressive strength of the mine wall $\sigma_p = 104.5$ Mpa, the thickness of the ore body is about 20–50 m. and the weighted average thickness is about 40 m. For the convenience of calculation, we equivocate it with a model with a height of 40 m, therefore $h = 40$ m. Furthermore, the mechanical parameters of the mine wall and the overlying surrounding rock are shown in Table 1.

**Table 1.** Mechanical parameters of mine wall and overlying surrounding rock.

| Name | Bulk Density/KN·m$^{-3}$ | Poisson | Cohesion/Mpa | Friction/° |
|------|--------------------------|---------|--------------|------------|
| Roof rock | 27.1 | 0.2087 | 36.50 | 35.3 |
| Mine wall | 32.2 | 0.3124 | 21.43 | 50.21 |

The above four factors are analyzed by the orthogonal range analysis method, combined with the actual situation of the Dongguashan project. Each factor is assigned within an appropriate range, and the calculation results of the mine wall safety factor under the conditions of each mine wall mining design scheme are shown in Table 2. The stability coefficient range results are shown in Table 3.

From the orthogonal test results in Table 2, it can be seen that the comprehensive safety factor of the mine wall is mostly the compression safety factor or the overturning safety factor. The main form of wall instability is brittle compression failure or overturning failure. According to the extreme difference of each factor in Table 3, the important factors affecting the stability of the mine wall are the width of the mine wall, the bulk density of the backfill, the friction angle of the backfill, and the width of the mine room. In order to further study the variation law between the factors affecting mine wall stability and the

mine wall safety factor, the research factors are regarded as variables and other factors are regarded as quantitative, and the quantitative relationship between each factor and the mine wall comprehensive safety factor is studied by the control variable method.

**Table 2.** Orthogonal test table of mine wall safety factor.

| Test Number | $l_1$/m | $b$/m | $\varphi$/° | $\gamma_1$/KN·m$^{-3}$ | $K_\sigma$ | $K_\tau$ | $K_c$ | $K$ |
|---|---|---|---|---|---|---|---|---|
| 1 | 60 | 2 | 20 | 20 | 0.3870 | 1.9867 | 0.7080 | 0.6863 |
| 2 | 60 | 3 | 28 | 32 | 0.7225 | 2.5390 | 0.8276 | 0.7225 |
| 3 | 60 | 4 | 36 | 24 | 1.9760 | 4.9425 | 2.2466 | 1.9760 |
| 4 | 60 | 5 | 24 | 36 | 1.5123 | 2.7669 | 1.0465 | 1.0465 |
| 5 | 60 | 6 | 32 | 28 | 3.2470 | 4.9980 | 2.4223 | 2.4223 |
| 6 | 65 | 2 | 36 | 32 | 0.4456 | 2.6540 | 0.6855 | 0.4456 |
| 7 | 65 | 3 | 24 | 24 | 0.8124 | 2.3587 | 0.9161 | 0.8124 |
| 8 | 65 | 4 | 32 | 36 | 1.2796 | 3.0240 | 1.0286 | 1.0286 |
| 9 | 65 | 5 | 20 | 28 | 1.6343 | 2.5561 | 1.1373 | 1.1373 |
| 10 | 65 | 6 | 28 | 20 | 3.6705 | 4.7018 | 2.7559 | 2.7559 |
| 11 | 70 | 2 | 32 | 24 | 0.4952 | 2.3765 | 0.7222 | 0.4952 |
| 12 | 70 | 3 | 20 | 36 | 0.5020 | 1.4942 | 0.4386 | 0.4386 |
| 13 | 70 | 4 | 28 | 28 | 1.3959 | 2.7750 | 1.0948 | 1.0948 |
| 14 | 70 | 5 | 36 | 20 | 3.3561 | 5.2323 | 2.7465 | 2.7465 |
| 15 | 70 | 6 | 24 | 32 | 2.3529 | 2.9248 | 1.2510 | 1.2510 |
| 16 | 75 | 2 | 28 | 36 | 0.3050 | 1.5236 | 0.3536 | 0.3050 |
| 17 | 75 | 3 | 36 | 28 | 1.0779 | 2.9706 | 0.9955 | 0.9955 |
| 18 | 75 | 4 | 24 | 20 | 1.6238 | 2.5372 | 1.1667 | 1.1667 |
| 19 | 75 | 5 | 32 | 32 | 2.1605 | 3.2314 | 1.2491 | 1.2491 |
| 20 | 75 | 6 | 20 | 24 | 2.6039 | 2.6640 | 1.3099 | 1.3099 |
| 21 | 80 | 2 | 24 | 28 | 0.3370 | 1.3522 | 0.3552 | 0.3370 |
| 22 | 80 | 3 | 32 | 20 | 1.2449 | 2.6173 | 1.0063 | 1.0063 |
| 23 | 80 | 4 | 20 | 32 | 0.9804 | 1.6180 | 0.5517 | 0.5517 |
| 24 | 80 | 5 | 28 | 24 | 2.4243 | 2.9227 | 1.2762 | 1.2762 |
| 25 | 80 | 6 | 36 | 36 | 3.2303 | 3.8513 | 1.4194 | 1.4194 |

**Table 3.** Stability coefficient range analysis results.

| The Mean of $K$ | $l_1$/m | $b$/m | $\theta$/° | $\gamma_1$/KN·m$^{-3}$ |
|---|---|---|---|---|
| $K_1$ | 1.3707 | 0.4538 | 0.8248 | 1.6723 |
| $K_2$ | 1.2360 | 0.7951 | 0.9227 | 1.1739 |
| $K_3$ | 1.2052 | 1.1636 | 1.2309 | 1.1974 |
| $K_4$ | 1.0052 | 1.4911 | 1.2403 | 0.8440 |
| $K_5$ | 0.9181 | 1.8317 | 1.5166 | 0.8476 |
| R | 0.4526 | 1.3779 | 0.6918 | 0.8284 |

*4.1. Quantitative Relationship between Mine Wall Safety Factor and the Width of the Mine Wall*

Other factors affecting the safety factor of the mine wall are regarded as fixed values, and study the relationship between the safety factor of the mine wall and the width of the mine wall, the width of the mine room $l_1 = 70$, the friction angle of the backfill $\theta = 28°$, and the bulk density of the backfill $\gamma_1 = 24$ KN·m$^{-3}$. Figure 3 is a diagram showing the relationship between the safety factor and the width of the mine wall.

As shown in Figure 3, with the increase of the width of the mine wall, the bearing strength and the anti-overturning capacity of the mine wall will increase. Therefore, the mine wall compression safety factor, shear safety factor and overturn safety factor all increase with the increase of the width of the mine wall. Analysis of the change rate of the safety factor shows that with the increase of the width of the mine wall, the increase rate of the shear safety factor and the overturning safety factor is basically stable, and the increase rate of the compression safety factor increases, which indicates that the width of the mine wall has the most significant effect on the compression safety factor. Under the value level of the above factors, when the width of the mine wall is less than 3 m, the mine



wall comprehensive safety factor is less than 1, the mine wall is in an unstable state, and the main failure forms are compression failure and overturning failure. With the increase in the width of the mine wall, the mine wall compression safety factor increases sharply. When the width of the mine wall is greater than 4 m, the mine wall safety factor is greater than 1, the mine wall compression safety factor is greater than the overturning safety factor, and the backfill has a more significant effect on the mine wall bending.

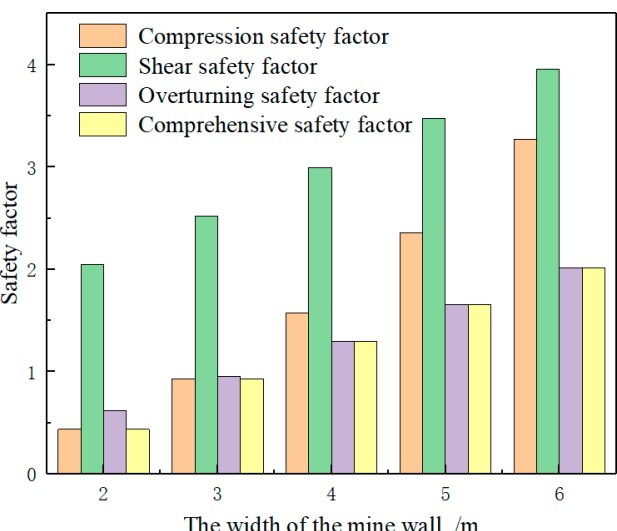

**Figure 3.** Relationship between safety factor and the width of the mine wall.

*4.2. Quantitative Relationship between Safety Factor of Mine Wall and Bulk Density of the Backfill*

The width of the mine wall is taken as 4 m, the width of the mine room is taken as 70 m, and the friction angle of the backfill is taken as 28°. The quantitative relationship between the safety factor of the mine wall and the bulk density of the backfill is studied. The results are shown in Figure 4.

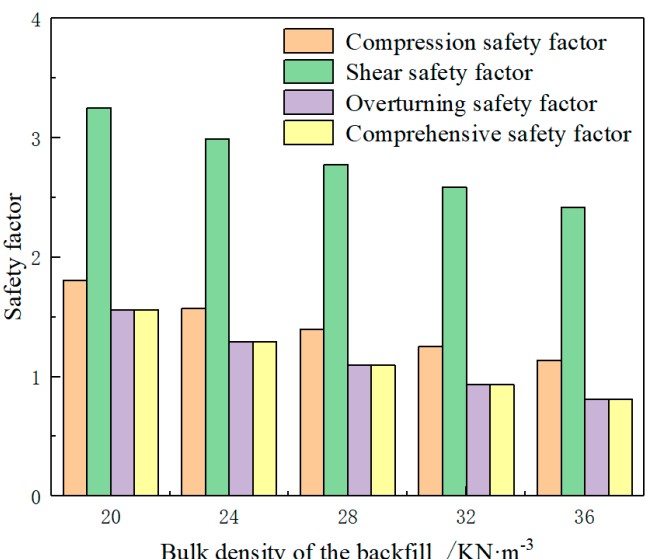

**Figure 4.** Relationship between safety factor and bulk density of the backfill.

It can be seen from Figure 4 that the mine wall compression safety factor, shear safety factor and overturning safety factor all decrease with the increase of the bulk density of the backfill, but the decreasing rate of all three decreases gradually. Comparing the mine wall compression safety factor and the change rate of the overturning safety factor, we can see that the overturning safety factor of the mine wall changes faster with the bulk

density of the backfill. Therefore, when considering the influence of the bulk density of the backfill on the stability of the mine wall, the overturning failure form of the mine wall should be considered.

### 4.3. Quantitative Relationship between the Mine Wall Safety Factor and the Friction of the Backfill

Controlling other variables, the variation law of the mine wall safety factor with the friction angle of the backfill is analyzed. The width of the mine wall and the width of the mine room are 4 m and 70 m, respectively, and the bulk density of the backfill is 24 KN·m$^{-3}$. Figure 5 is a diagram showing the relationship between the mine wall safety factor and the friction angle of the backfill.

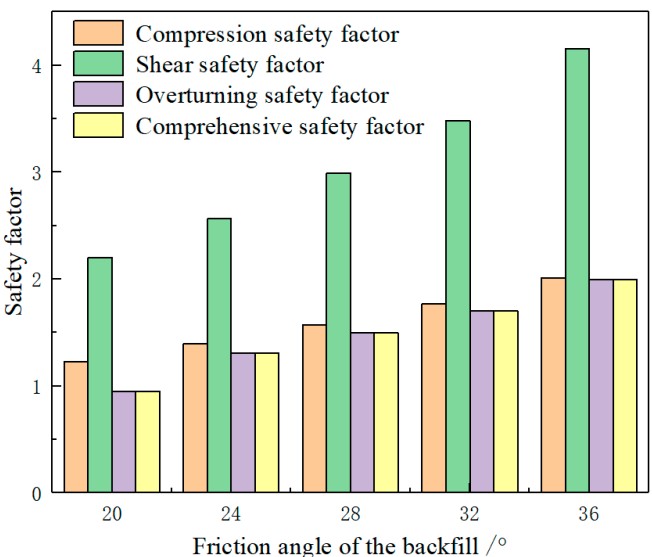

**Figure 5.** Relationship between safety factor and friction angle of the backfill.

It can be seen from Figure 5 that with the increase of the friction angle of the backfill, the safety factor of the mine wall increases, and the increase rate of the compression safety factor remains basically unchanged, while the increase rate of the shear safety factor and the overturning safety factor gradually increases. The analysis shows that when the friction angle of the backfill increases, the lateral pressure coefficient decreases, the effect of the backfill on the mine wall decreases, and the safety factor of the mine wall increases accordingly. Because the friction angle in the backfill mainly affects the side of the backfill on the mine wall. Therefore, the shear safety factor and the overturning safety factor are more affected by the friction angle of the backfill.

### 4.4. Quantitative Relationship between Mine Wall Safety Factor and the Width of the Mine Room

The width of the mine wall is 4 m, the bulk density and friction angle of the backfill are 24 KN·m$^{-3}$ and 28°, respectively, and the variation law of the safety factor of the mine wall with the width of the mine room is studied. The results are shown in Figure 6.

With the increase of the width of the mine room, the load of the overlying surrounding rock on the backfill increases, the load on the overlying surrounding rock on the mine wall decreases, and the lateral load of the backfill increases. Therefore, with the increase in the width of the mine room, the compression safety factor of the mine wall increases gradually and the shear safety factor and overturning safety factor of the mine wall both decrease, which is consistent with the trend shown in Figure 6. At the level of the above factors, when the width of the mine room exceeds 80 m, the mine wall comprehensive safety factor is taken as the overturning safety factor and the mine wall is more prone to overturning failure.

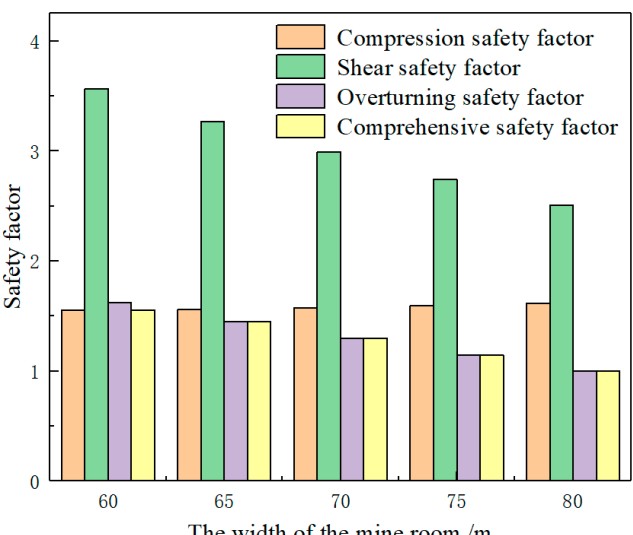

**Figure 6.** Relationship between safety factor and the width of the mine room.

*4.5. Multi-Factor Analysis of Mine Wall Safety Factor*

In order to comprehensively analyze the variation law of the comprehensive safety factor of the mine wall with the width of the mine wall, the width of the mine room, the bulk density of the backfill and the friction angle of the backfill, a regression equation of the safety factor including four factors was established, and the quantitative relationship is used to quantitatively reflect the change of the comprehensive safety factor of the mine wall. According to the orthogonal experimental calculation results in Table 2, a multiple regression equation is constructed by the Matlab data processing software and the expression of the regression equation can be obtained as:

$$y = 1.5513 - 0.0227x_1 + 0.3452x_2 + 0.0425x_3 - 0.0495x_4 + \varepsilon \tag{16}$$

In Equation (16): the dependent variable $y$ represents the comprehensive safety factor of the mine wall; $x_1$ represents the width of the mine room (m); $x_2$ represents the width of the mine wall (m); $x_3$ represents the friction angle of the backfill (°); $x_4$ represents the bulk density of the backfill (KN·m$^{-3}$); $\varepsilon$ represents the residual.

The complex correlation coefficient of the regression equation in Equation (16) $R = 0.8848$ shows that the regression equation fits well, and $F = 38.4076$, $p = 0.0000$ shows that the explanatory variable ($x_i$) is significant for the coefficient test. Figure 7 is the residual of the regression equation. It can be seen that there is only one abnormal point, which also shows that the regression effect is good.

According to the independent variable coefficient sign of the regression equation of the comprehensive safety factor, the comprehensive safety factor of the mine wall changes in the same direction as the width of the mine wall and the friction angle of the backfill. The increase in the width of the mine wall or the friction angle of the backfill will increase the comprehensive safety factor of the mine wall. The comprehensive safety factor of the mine wall changes inversely with the width of the mine room and the bulk density of the backfill, which is the same as the above quantitative analysis results. According to the size of the independent variable coefficient of the expression of the regression equation, the important factors affecting the stability of the mine wall are: the width of the mine wall, the bulk density of the backfill, the friction angle of the backfill, and the width of the mine room, which are the same as those shown in Table 3. At the same time, according to the independent variable coefficient value of the expression of the regression equation, the influence weight of each factor on the comprehensive safety factor of the mine wall can be quantitatively analyzed.

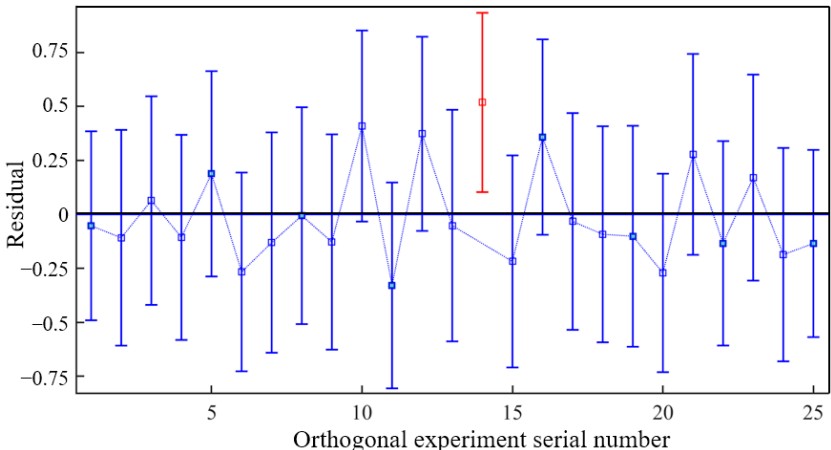

**Figure 7.** Residual diagram of the regression equation of the comprehensive safety factor of the mine wall.

## 5. Engineering Examples

According to the design scheme of isolated pillar mining in the Dongguashan Copper Mine, the width of the mine wall is 4 m, the width of the mine room is 78 m, and the filling scheme of the mine room uses full tailings cement filling. Experiments show that the bulk density of the backfill is 24 KN·m$^{-3}$ and the friction angle is about 34.6°. According to Equation (15), the comprehensive safety factor of the mine wall is 1.39, and the comprehensive safety factor of the mine wall calculated by the regression equation of Equation (16) is 1.46; the error rate is less than 6%. Furthermore, the comprehensive safety factors of the mine wall obtained by the two equations both exceed the critical safety factor of 1.00. Theoretical calculation shows that under this mining design scheme, the mine wall is in a stable state.

According to the above design scheme, the isolated pillar is mined. After the mining is completed, the CMS goaf detection technology is used to scan the pillar back to the mined area to observe the retention of the mine wall [35], and use the BGK-A3 displacement meter to monitor the deformation of the roof rock during the mining process [36]. The triaxial stress meter is used to monitor the stress change of the mine wall. Figure 8 is a cross-sectional view of the scanning results of the open area after the mining of the 52-line isolated pillar G5 is completed.

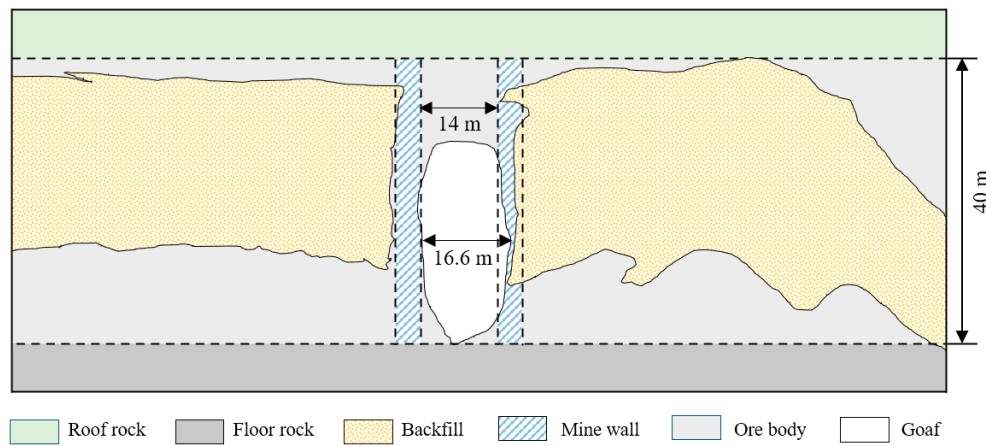

**Figure 8.** Vertical cross section view of empty area scan results.

It can be seen from Figure 8 that after the G5 isolation pillar is mined, the integrity of the top of the mine wall remains relatively good, no overturning failure occurs, no fracture occurs at the bottom of the mine wall and the pressure bearing performance is good. The integrity of the left mine wall is good, but there is over-mining in the middle of the right

mine wall. The maximum span of the goaf is 16.6 m, and there is no large amount of backfill mixed in the gob scanning and ore mining. The maximum displacement of the roof surrounding rock measured by the BGK-A3 displacement meter is only 32 mm, and the deformation is in a controllable range. During the mining of the isolated pillar, the relative change of the maximum principal stress of the mine wall is 1.16 Mpa, the change of the intermediate principal stress is 1.17 Mpa, and the minimum change of the principal stress is 2.05 Mpa. The change in the stress value of the mine wall is small and it is in a stable state. The reliability of the mine wall is good, which is consistent with the theoretical calculation results and meets the mining requirements. In addition, according to the existing research results [15,37,38], this article analyzes the stability of the mine wall under this parameter by the energy variation method, the cusp catastrophe theory and the thin plate theory. The calculation results are consistent with the above situation, which further verifies the theoretical analysis results and improves the reliability of the research results.

## 6. Conclusions

In this study, according to the mining situation of an isolated pillar in the Dongguashan Copper Mine, the bearing mechanism of the backfill–mine wall system is analyzed, and the influence of backfill mechanical parameters on the stability of the mine wall is considered. Based on the failure forms of the mine wall under multi-directional loads, the comprehensive evaluation index of the mine wall stability is proposed, the calculation equation of the comprehensive safety factor is deduced, the orthogonal test is designed to analyze the sensitivity of the influencing factors of the mine wall stability, and the following research results are obtained:

1. The important factors affecting the stability of the mine wall are the width of the mine wall, the bulk density of the backfill, the friction angle of the filling body and the width of the mine room. Among them, the width of the mine wall mainly affects the bearing strength and the stress distribution state of the mine wall. The bulk density of the backfill and the friction angle mainly affect the lateral load of the mine wall, and the width of the mine room affects both the vertical load and the lateral load of the mine wall.

2. The main forms of mine wall failure are brittle compression failure and overturning failure. Increasing the width of the mine wall can significantly improve the mine wall compression safety factor. Reducing the bulk density of the backfill and increasing the friction angle of the backfill can improve the mine wall overturning safety factor. The increase of the mine width increases the compression safety factor and reduces the overturning safety factor of the mine wall.

3. According to the regression equation calculation, the comprehensive safety factor of the 52-line G5 mine wall is about 1.4, which is close to the theoretical equation and the actual situation of engineering exploration. It provides ideas for the optimization of mine wall design and filling scheme in the process of deep isolation pillar mining.

**Author Contributions:** Conceptualisation, J.G. and J.L.; methodology, X.C.; validation, X.X.; writing—original draft preparation, X.C. and Y.Z.; writing—review and editing, X.C. and Y.Z. All authors have read and agreed to the published version of the manuscript.

**Funding:** Project supported by the National Natural Science Foundation of China Received (No. 52174140); Project supported by the National Natural Science Foundation of China Received (No. 52074351); Postgraduate Scientific Research Innovation Project of Hunan Province (No. QL20210054); Supported by Postgraduate Innovative Project of Central South University "Research and practice of comprehensive reinforcement technology of wellbore in water-rich broken soil layer" (No. 2021XQLH066).

**Acknowledgments:** The team of authors express their gratitude to the editors and reviewers for valuable recommendations that have been taken into account to improve significantly the quality of this article.

**Conflicts of Interest:** The authors declare no conflict of interest.

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
