# Peer review of "Research on Factors Affecting Mine Wall Stability in Isolated Pillar Mining in Deep Mines"

_minerals, doi:10.3390/min12050623_

Round 1

Reviewer 1 Report

This paper proposes an evaluation index and computations are derived for a pillar in underground mine. This is based on mechanical model under trapezoidal load distribution due to backfill. The paper needs to be improved before it can be processed further. See my comments attached.

Author Response

Thank you for taking time out of your busy schedule to review the manuscript. We have carefully corrected and responded to the manuscript as you suggested. Modification instructions are attached. If you have any questions, please don't hesitate to contact us, thank you. Please see attachment.

Reviewer 2 Report

The article is very well presented with the rigorous quality of the scientific research process. I have some questions that the authors could support with references or theoretical arguments:

How do you determine the value of α in equation (8)?

How do you determine the value of c in equation (10)?

In Figure 2b, considering that the lower end of the wall is fixed and is located at a distance h from the applied load p, it could be assumed that, due to the product of the load by the distance h, at the ends higher, the greater the distance from the lower end, the greater lateral load is exerted; In this context, could you explain in your article why in your model the lateral load is higher at the top and lower at the bottom?

The development and discussion of the results are presented in a coherent manner and the conclusions are supported by the results and the discussion in the article.

The list of references are sufficient and current.

  1. The main question addressed by the investigation is an analysis of the support of the backfill mine wall system, which is based on the failure forms of the mine wall under multidirectional loads, through the comprehensive evaluation index of the stability of the backfill wall. For which the authors propose an equation for calculating the integral security factor based on a physical model.
  2. The investigation is interesting because of the way of approaching the problem and obtaining a proven solution; and it is also relevant from the point of view that the results provide information that could be reproduced for other mines or similar problems.
  3. The originality of the topic is acceptable, since the authors use an analytical perspective of the problem to solve it with practical results.
  4. The addition to the thematic area is the perspective of analysis of the support mechanism of the fill wall after the extraction of the isolated pillar, with which the authors deduce an expression of the safety factor of the mine wall under the action of the strata overlying and fill, and with it they design orthogonal experiments to study the influence of different parameters on the stability of the mine wall. While other authors have retained a kind of mine wall stability and reliability analysis mainly on one-way vertical loading, and ignored the influence of backfill on mine wall stability.
  5. The text is written in a legible format and suitable for understanding, the conclusions are consistent with the results and the theoretical arguments explained.
  6. The work presented clearly responds to the problem raised, and is supported by scientific evidence.

Author Response

(The authors gave the same response as above.)

Reviewer 3 Report

In the presented paper taking Dongguashan Copper Mine as an example, the authors made the analysis of the bearing mechanism of backfill-mine wall system. They showed that the width of the mine wall stromgly affects the bearing strength and the stress distribution state of the mine wall, while the bulk density of the backfill and the friction angle mainly affect the lateral load of the mine wall.

I beleive that the obtained results will be interesting to thereaders of Minerals.

Author Response

(The authors gave the same response as above.)
